# Negative Thermal Expansion Properties of $Sm_{0.85}Sr_{0.15}MnO_{3-\delta}$

**Yucheng Li *, Yang Zhang, Yongtian Li and Yifeng Wu**

Department of Avionics Engineering, Aviation Maintenance NCO Academy, Air Force Engineering University, Xinyang 464000, China; zy9804020@163.com (Y.Z.); 13937616646@139.com (Y.L.); 13937616450@139.com (Y.W.)
* Correspondence: zzuleeyucheng@163.com; Tel./Fax: +86-376-6655-656

**Abstract:** A novel negative thermal expansion (NTE) material composed of $Sm_{0.85}Sr_{0.15}MnO_{3-\delta}$ was synthesized using the solid-state method. By allowing $Sr^{2+}$ to partially replace $Sm^{3+}$ in $SmMnO_3$, the ceramic material $Sm_{0.85}Sr_{0.15}MnO_{3-\delta}$ exhibits NTE properties between 360K and 873K, and its average negative thermal expansion coefficient was $-10.08 \times 10^{-6}$/K. The structure of $Sm_{0.85}Sr_{0.15}MnO_{3-\delta}$ is orthogonal, the space group is pbnm, the morphology is regular, and the grain size is uniform. The results of X-ray diffraction and XPS (X-ray photoelectron spectroscopy) suggest that the NTE phenomenon is related to the electron transfer of Mn ions. With the increase in temperature, $Mn^{4+}$ is rapidly transformed into $Mn^{3+}$, accompanied by $Mn^{4+}O_6$ octahedron distortion and oxygen defects. It was found that the sample volume continually decreased at the same time.

**Keywords:** negative thermal expansion; $Sm_{0.85}Sr_{0.15}MnO_{3-\delta}$; lattice distortion; oxygen defects; Jahn–Teller effect

## 1. Introduction

We know that most instruments are composed of various materials, but with increases in temperature, different thermal expansion coefficients of various constituent materials may lead to thermal mismatches, and small cracks in the equipment can lead to performance failures and even instrument damage. NTE materials have attracted considerable research attention in the production of composites with accurately controllable positive, negative, or zero coefficients of thermal expansion [1–11].

A great number of NTE materials have been found, such as oxides ($Cu_{1.5}Mg_{0.5}V_2O_7$, $Cu_2V_2O_7$, and $HfMnMo_3O_{12}$, etc.) [12–16], antiperovskite $Mn_3XN$, and perovskite ($BiNiO_3$, $Gd_{1-x}Sr_xMnO_{3-\sigma}$, and $Er_{0.7}Sr_{0.3}NiO_{3-\delta}$, etc.) [17–21]. However, each material has limitations because of some defects. $ZrW_2O_8$ is a metastable phase at room temperature (RT), which is difficult to prepare due to it readily decomposing [1]. $ZrV_2O_7$ exists as a phase transformation at 375K [2]. $Y_2Mo_3O_{12}$ has a water-absorbing quality at RT. Although antiperovskite ($Mn_3Cu(Ge)N$, $Mn_3NiN$, and $Mn_3ZnN$, etc.) possesses the properties of superconductivity, giant magnetoresistance, magnetocaloric effects, and constant electrical resistivity [8], the NTE temperature range is usually under RT, and its preparation conditions are very strict. $Mn_3Cu(Ge)N$ needs to be grown on a silicon surface with high pressure and argon gas protection. The NTE perovskite $ABO_3$ (A = Gd, Er, and Bi, etc.; B = Mn, Er, Sr, Ni and Sr, etc.) not only shows NTE properties in a large temperature range above RT but also has simple preparation conditions.

Kurimamachiya-chouses conducted research on $Sr^{2+}$ partly substituting $Gd^{3+}$ in $GdMnO_3$. They pointed out that $Gd_{1-x}Sr_xMnO_{3-\delta}$ had excellent NTE properties [18]. L. J. Fu reported the NTE material of $Er_{0.7}Sr_{0.3}NiO_{3-\delta}$ with $Sr^{2+}$ partly substituting $Er^{3+}$ in $ErNiO_3$ [19]. These studies suggest that the substituting method is an effective way to prepare new kinds of NTE materials with excellent properties [16,18–20]. In the present study, we conducted research on $Sr^{2+}$ partly substituting $Sm^{3+}$ in $SmMnO_3$. The thermal properties are discussed.

## 2. Experimental Procedures

The sample was prepared according to the conventional solid-state method. Analytic-grade $Sm_2O_3$ (purity 99.5%), SrO (purity 99.5%), and $MnO_2$ powder were used as raw materials. Using $MnO_2$ as the raw material, $Mn_2O_3$ powder was prepared by burning in a 923 K furnace for 10 h.

$Sm_2O_3$, SrO, and $Mn_2O_3$ powders were mixed according to the mole ratio of Sm:Sr:Mn = 0.85:0.15:1. The mixtures were ground using an agate mortar for 1 h and then ground with ethanol for 2 h. The obtained mixtures were then dried for 1 h at 353 K in a baking oven. Afterward, the mixtures were pressed into cylindrical-shape compacts (Ø10 × 5 mm) using a powder pellet machine (769YP-15A, 200 MPa). The compacts were initially sintered in a pipe furnace (AY-BF-555-180) at 1273 K for 10 h in air and subsequently sintered at 1623 K for 10 h. The sample was allowed to cool in the furnace naturally.

The linear thermal expansion coefficient was measured using a dilatometer Linseis L76 (heating and cooling rates of 5 K/min). The XRD measurement was carried out using Bruker D8 Advance with CuKα radiation. The XRD pattern of the sample was analyzed using X'Pert HighScore Plus software. The lattice constants a, b, and c and the unit cell volume of the sample were calculated using powderX software and the least square method. The surface morphology of the sample was observed using the FEI Quanta 250 scanning electron microscopy (SEM), and the EDS energy spectrum was obtained using an Appllo XP. The TGA and DSC were tested using a LabsysTM thermal analyzer. The XPS (X-ray photoelectron spectroscopy) was performed using a Thermo Scientific K-Alpha instrument for the valence analysis of the Mn element. The BET tests were performed to determine the size and volume of the holes using an ASAP2460 device.

## 3. Results and Discussion

### 3.1. Phase Analysis

Figure 1a is the XRD pattern of the sample at RT. Comparing the XRD pattern with the JCPDS cards for $SmMnO_3$ (00-025-0747), $Eu_{0.9}Sr_{0.1}MnO_3$ (No. 00-051-0252), and $Eu_{0.8}Sr_{0.2}MnO_3$ (00-051-0251), we found that the diffraction peaks were similar to those of the JCPDS cards, except for some shifts, which suggests that the as-prepared sample had similar structure to that of $SmMnO_3$, $Eu_{0.9}Sr_{0.1}MnO_3$, and $Eu_{0.8}Sr_{0.2}MnO_3$. It can be confirmed that the ceramic $Sm_{0.85}Sr_{0.15}MnO_{3-\delta}$ crystallizes in an orthorhombic structure. As the ionic radius of $Sr^{2+}$ (ionic radius 1.18 Å) is bigger than that of $Sm^{3+}$ (ionic radius 0.958 Å), the difference in the ionic radius may cause lattice distortion. As $Sr^{2+}$ partly substitutes for $Sm^{3+}$, the diffraction peaks also shift.

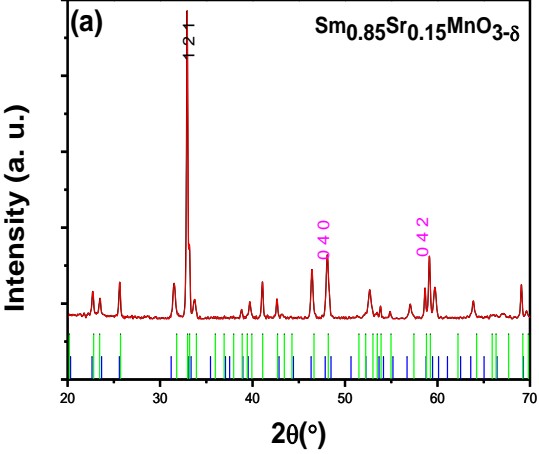 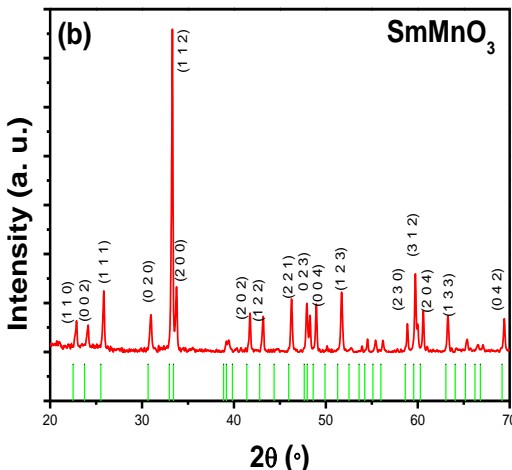

**Figure 1.** The XRD patterns: (**a**) $Sm_{0.85}Sr_{0.15}MnO_{3-\delta}$ and (**b**) $SmMnO_3$.

Figure 2a shows the SEM image of the sample. We found that the ceramic sample was composed of homogenous spherical or elliptic spherical particles with some obvious agglomerations. There were pores and microcracks in the sintered body. The size of the particles was uniform, with an average grain size of about 1~2 μm. The EDS analysis of the sample revealed the primary elements of Sm, Sr, Mn, and O, and their atomic ratio (Sm:Sr:Mn:O) was about 0.85:0.15:1:3 (seeing Table 1). Combined with the XRD analysis, we identified the composition of the samples as being $Sm_{0.85}Sr_{0.15}MnO_3$.

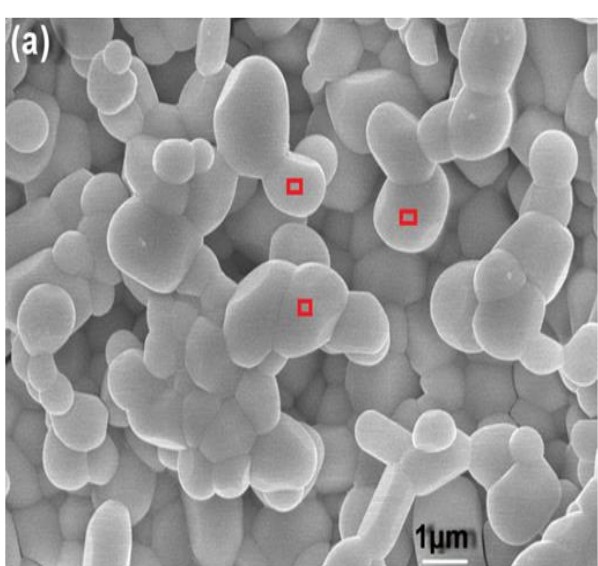 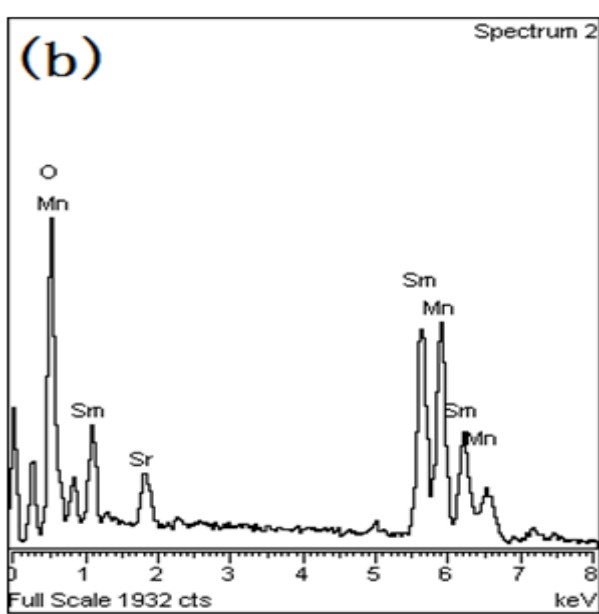

**Figure 2.** (**a**) SEM image of the ceramic $Sm_{0.85}Sr_{0.15}MnO_{3-\delta}$; (**b**) EDS spectrum corresponding to the SEM image.

**Table 1.** Atomic ratio of Sm, Sr, Mn, and O in $Sm_{0.85}Sr_{0.15}MnO_{3-\delta}$ by EDS.

| Element | Sm | Sr | Mn | O |
|---|---|---|---|---|
| (at.%) | 14.46 | 2.39 | 16.20 | 66.95 |

### 3.2. Thermal Expansion Property

Figure 3a–c show the relative length (**dL/L**) with the temperature increases of $SmMnO_3$, $SrMnO_3$, and $Sm_{0.85}Sr_{0.15}MnO_{3-\delta}$, respectively. $SmMnO_3$ (Figure 3a) and $SrMnO_3$ (Figure 3b) showed positive thermal expansion. Calculating according to the curve, the expansion coefficients were $5.24 \times 10^{-6}$/K and $12.7 \times 10^{-6}$/K, respectively. When the temperature was below 360 K, the ceramic $Sm_{0.85}Sr_{0.15}MnO_{3-\delta}$ showed a positive thermal expansion of $0.46875 \times 10^{-6}$/K. As the temperature increased, the ceramic $Sm_{0.85}Sr_{0.15}MnO_{3-\delta}$ showed an NTE property in the range of 360 to 873 K. The average linear expansion coefficient was $-10.08 \times 10^{-6}$/K.

Figure 4 shows the high-temperature XRD patterns of ceramic $Sm_{0.85}Sr_{0.15}MnO_{3-\delta}$ from RT to 873 K. As the temperature increased, the diffraction peaks of $Sm_{0.85}Sr_{0.15}MnO_{3-\delta}$ moved slightly to small angles, except three diffraction peaks (31.54°, 33.79°, and 52.65°) that moved to a large angle.

Figure 5 shows the variation in the $Sm_{0.85}Sr_{0.15}MnO_{3-\delta}$ lattice parameters and volume with temperature increases, which was calculated using the powderX software. In a, c in Figure 5, the increase occurred gradually, while in b in Figure 5, it decreased as the temperature increased gradually. We believe that the thermal expansion of $Sm_{0.85}Sr_{0.15}MnO_{3-\delta}$ was due to anisotropy. We can see that from RT to 360 K, $Sm_{0.85}Sr_{0.15}MnO_{3-\delta}$ showed a positive expansion property. As the temperature increased to 360~873 K, $Sm_{0.85}Sr_{0.15}MnO_{3-\delta}$ showed an

NTE property with the average linear expansion coefficient of $-3.33 \times 10^{-6}$/K. However, the original calculation of the negative thermal expansion coefficient of $Sm_{0.85}Sr_{0.15}MnO_{3-\delta}$ in this temperature range was $-10.08 \times 10^{-6}$/K, according to Figure 3. As can be seen from Figure 2 above, there were pores and microcracks in the crystal. Therefore, we believe that when the temperature rises, the crystal squeezes the open space, namely, these pores and microcracks, which is another reason for the negative thermal expansion.

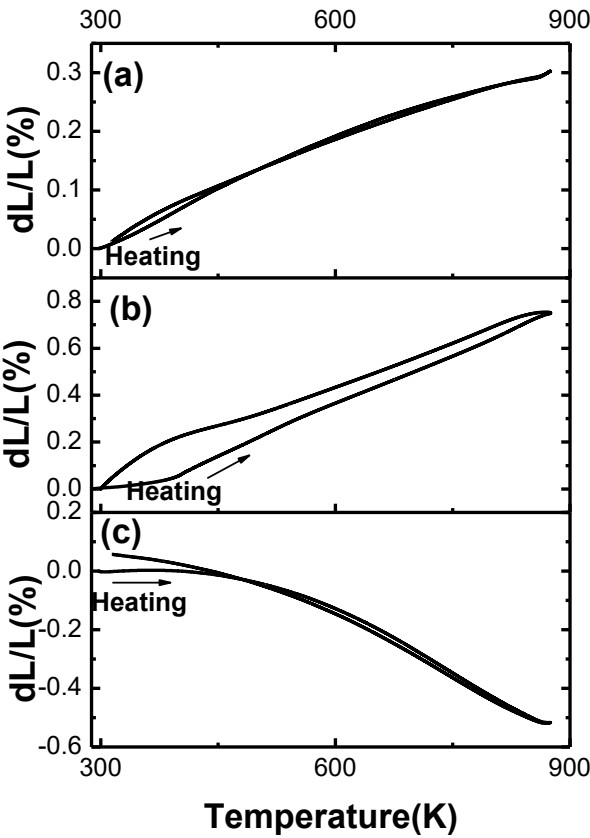

**Figure 3.** Relative length change (*dL/L*) with the temperature of the samples: (**a**) $SmMnO_3$, (**b**) $SrMnO_3$, and (**c**) $Sm_{0.85}Sr_{0.15}MnO_{3-\delta}$.

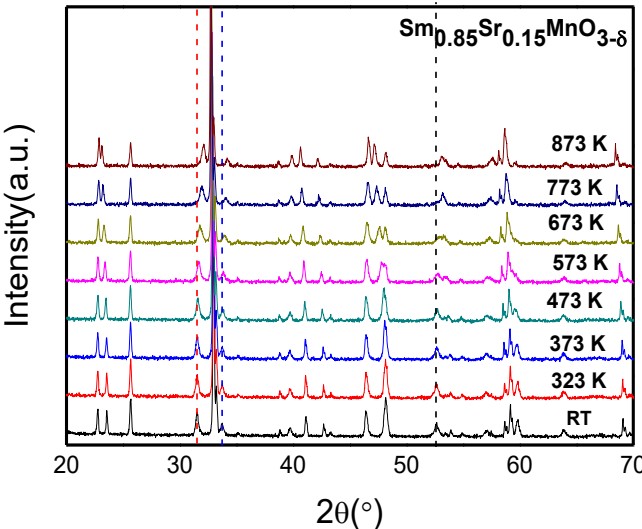

**Figure 4.** XRD patterns of $Sm_{0.85}Sr_{0.15}MnO_{3-\delta}$ ceramics at high temperatures.

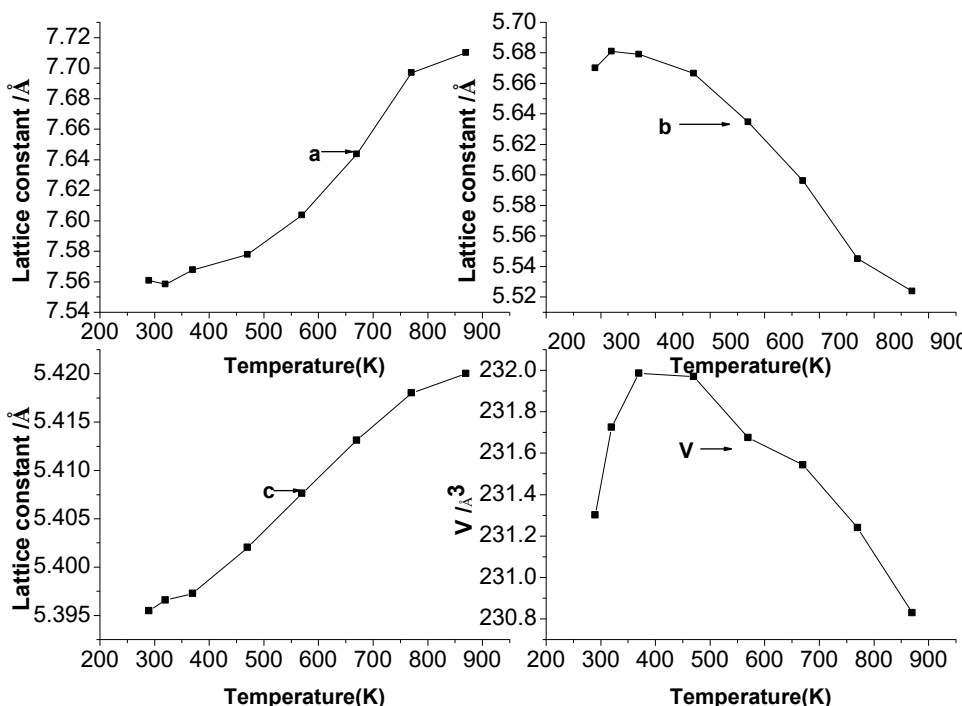

**Figure 5.** The variation in the $Sm_{0.85}Sr_{0.15}MnO_{3-\delta}$ lattice parameters and volume with temperature increases.

Table 2 shows the pore size, pore volume, and BET surface area of the sample. The specific surface of the material itself is large, and the general level of adsorption is good. When the pore structure of carbon materials is more complex, it is easy to have a flexible hole, and the pore size becomes larger after gas adsorption. With the doping of $Sr^{2+}$, oxygen defects are caused, and the gas is adsorbed in the pores. With the further doping of $Sr^{2+}$, the adsorption oxygen saturation does not change. With the increase in temperature, the gas is sintered out, the b-axis shrinks at the same time, and the pore size becomes smaller, resulting in the negative expansion property.

**Table 2.** Pore size, pore volume, and BET surface area of the sample.

| Sample | Pore Size (nm) | Pore Volume (cm³/g) | BET Surface Area (m²/g) |
|---|---|---|---|
| $Sm_{0.85}Sr_{0.15}MnO_{3-\delta}$ | 15.7842 | 0.002563 | 0.6351 |

Figure 6a is the XPS spectrum of $Sm_{0.85}Sr_{0.15}MnO_{3-\delta}$; the characteristic peaks of Sm, Sr, Mn, and O are shown in the figure, respectively. The surface of the sample was free from any pollutants, and element C was used for the calibration of the XPS atlas. Figure 6b,c show the XPS spectra of Mn. In the XPS spectrum, the sample had a bimodal structure, which indicates that the Mn elements on the sample surface existed in two forms: $Mn^{3+}$ and $Mn^{4+}$, which led to the oxygen vacancy. The presence of the oxygen vacancy facilitated the movement of electrons between $Mn^{4+}$ and $Mn^{3+}$. The oxygen vacancy also led to the shortening of the bond length of the Mn-O bond, which led to lattice distortion and generated internal stress; this reduced the bond angle of Mn-O-Mn and increased the double-exchange effect.

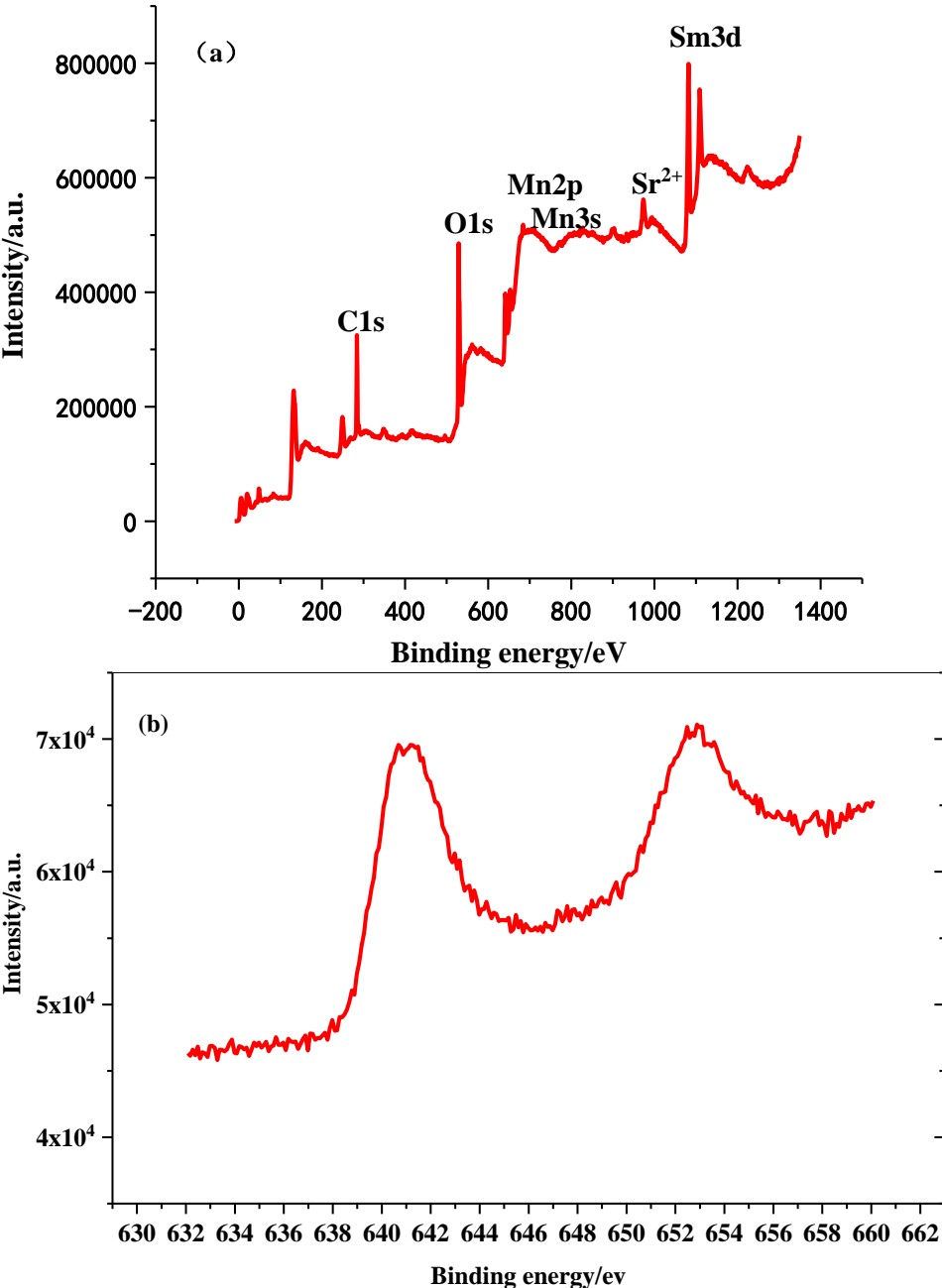

**Figure 6.** *Cont.*

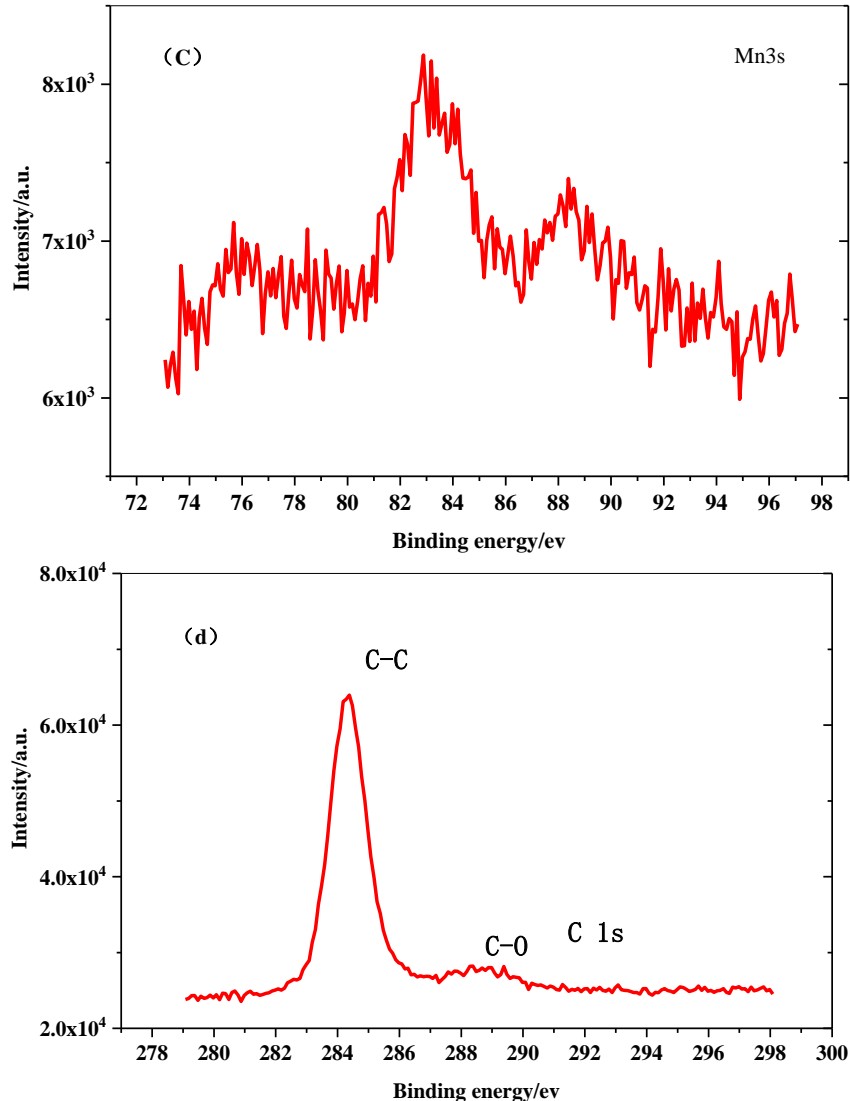

**Figure 6.** The XPS spectra of $Sm_{0.85}Sr_{0.15}MnO_{3-\delta}$: (**a**) elemental analysis, (**b**) Mn2p, (**c**) Mn3s, and (**d**) C1s.

### 3.3. Discussion

$SmMnO_3$ is a typical manganite perovskite structure. The structure of $SmMnO_3$ is shown in Figure 7. As for the $MnO_6$ octahedron in $SmMnO_3$, the distortion was caused by a change in the length of the Mn-O bond.

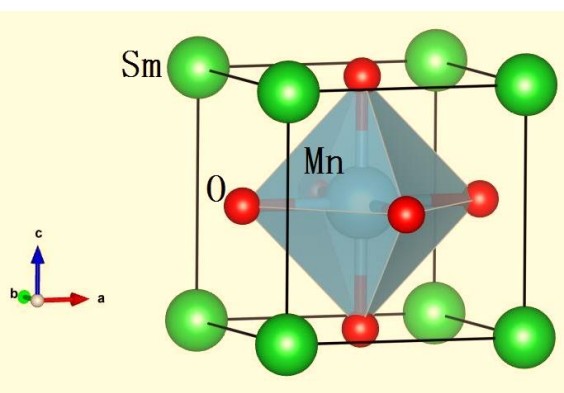

**Figure 7.** Structure diagram of $SmMnO_3$.

There are three kinds of common modes for this change [22–26] as follows. (1) The surface tension contract model $Q_1$, as shown in Figure 8a. Six oxygen atoms of the unit cell move close to or far away from the manganese atom at the same time, making the Mn-O bond length decrease or increase significantly. This model can increase the energy of the system, which is not conducive to the system energy being able to decrease and makes the system extremely unstable in turn. (2) The plane distortion model $Q_2$, as shown in Figure 8b. In a unit, two oxygen atoms in the horizontal plane leave a manganese atom, while the other two oxygen atoms become close to the manganese atom. The location of the two oxygen atoms in the vertical plane remains unchanged. (3) The expansion mode, or inspiratory mode $Q_3$, which is shown in Figure 8c. In a $MnO_6$ octahedron, the two oxygen atoms in the vertical plane leave manganese atoms, while the four oxygen atoms in the horizontal plane become close to the manganese atom simultaneously. For a $MnO_6$ octahedron, the $Q_1$ and $Q_2$ models normally exist. Since the $Q_1$ model is unstable, the distortion of the $MnO_6$ octahedron is mainly the $Q_2$ model, also called the plane distortion model.

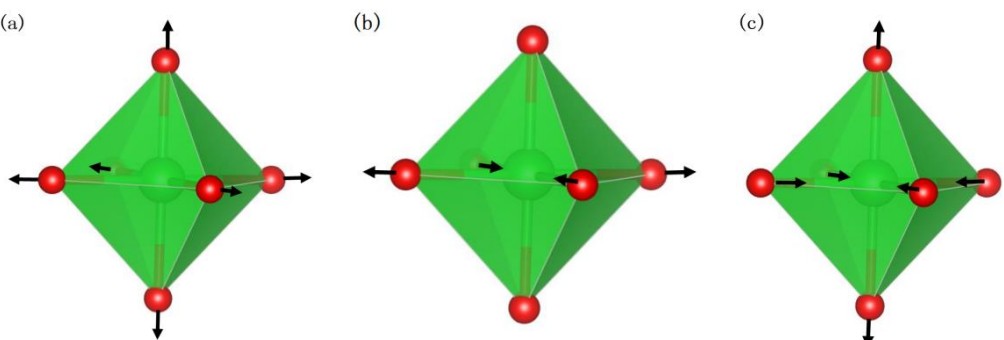

**Figure 8.** Three kinds of distortion models for the Mn-O bond (**a**) The surface tension contract model $Q_1$; (**b**) The plane distortion model $Q_2$; (**c**) The expansion mode, or inspiratory mode $Q_3$.

We used $MnO_2$, $Sm_2O_3$, and $SrO$ as the raw materials to prepare $Sm_{0.85}Sr_{0.15}MnO_{3-\delta}$. In the reaction process, there was a reciprocal transformation between $Mn^{3+}$ and $Mn^{4+}$.

When $Sr^{2+}$ substitutes the $Sm^{3+}$ in $SmMnO_3$, $Sr^{2+}$ will occupy the position of $Sm^{3+}$. To maintain the valence balance, electron transfer occurs in the $Mn^{3+}$ converting into $Mn^{4+}$ in $Sm_{0.85}Sr_{0.15}MnO_{3-\delta}$. Additionally, the p electron of $O^{2-}$ will migrate to the orbit of the nearby $Mn^{4+}$, and the d electron of $Mn^{3+}$ will migrate to the orbit of the nearby $Mn^{3+}$. Thus, this mechanism results in the electronic conduction and position exchanges of $Mn^{4+}$ and $Mn^{3+}$ ions. The system energy remains unchanged throughout. This process is known as the double exchange [27]. The structure of $Mn^{3+}$-$O^{2-}$-$Mn^{4+}$ forms in the process. However, according to the theory of Zener [28], the route of electron transfer between two $Mn^{3+}$ changes between $Mn^{3+}$ and $Mn^{4+}$. In order to keep the electron transfer between two $Mn^{3+}$, the magnetic moment between $Mn^{3+}$ and $Mn^{4+}$ ions should be parallel to each other. In this situation, it is favorable for there to be more electron transfer between $Mn^{3+}$ and $Mn^{4+}$ ions.

According to the analysis of the variable-temperature XRD data, we considered that the thermal property of $Sm_{0.85}Sr_{0.15}MnO_{3-\delta}$ might be related to the interaction of the lattice vibration and electron transfer between $Mn^{3+}$ and $Mn^{4+}$. As the temperature rose, the lattice vibrated dramatically and $Mn^{4+}$ converted into $Mn^{3+}$. Moreover, the electron transfer rate increased between the $Mn^{3+}$ and $Mn^{4+}$ ions. The number of $Mn^{3+}$ ions that can cause the Jahn–Teller [29] effect increased. The oxygen ions in the $Mn^{3+}O_6$ octahedron became slant, or even produced oxygen defects, making the unit cell volumes decrease. From RT to 360 K, the unit cell volume increased. The reason is that the contribution of the lattice vibration to the thermal expansion exceeded that of the $MnO_6$ octahedral distortion and oxygen defects. As the temperature increased, $Sm_{0.85}Sr_{0.15}MnO_{3-\delta}$ showed a low positive thermal expansion property, and above 360 K, the unit cell volume decreased. With more $Mn^{4+}$ ions converting into $Mn^{3+}$, the $Mn^{3+}O_6$ octahedral distortion was enhanced and oxygen defects occurred. These contributed more to the thermal expansion than the lattice

vibration. Therefore, $Sm_{0.85}Sr_{0.15}MnO_{3-\delta}$ shows a negative thermal expansion property between 360 K and 873 K.

The DSC and TGA results of ceramic $Sm_{0.85}Sr_{0.15}MnO_{3-\delta}$ also support the above statements. Figure 9a presents the DSC curve of ceramic $Sm_{0.85}Sr_{0.15}MnO_{3-\delta}$. In the curve, $Sm_{0.85}Sr_{0.15}MnO_{3-\delta}$ has an endothermic peak at about 360 K. This shows that more $Mn^{4+}$ ions were converted to $Mn^{3+}$ with the increase in temperature. Thus, $Mn^{3+}O_6$ octahedral distortion was enhanced and oxygen defects occurred. The unit cell volume began to decrease, which is consistent with the results calculated by the high-temperature XRD (seeing b in Figure 5). As electron transfer occurred between the $Mn^{3+}$ and $Mn^{4+}$ ions, the amount of $Mn^{4+}$ decreased, and oxygen ions in the $Mn^{3+}O_6$ octahedron became slant or even produced oxygen defects. The TGA results of $Sm_{0.85}Sr_{0.15}MnO_{3-\delta}$ confirm the existence of oxygen defects. In Figure 8b, the weight of the $Sm_{0.85}Sr_{0.15}MnO_{3-\delta}$ sample decreased when the temperature increased from RT to 873 K. In addition, the variable-temperature XRD (seeing Figure 5) showed that there was no phase transition with the increase in temperature. As electron transfer occurred between the $Mn^{3+}$ and $Mn^{4+}$ ions, $Mn^{4+}O_6$ converted into $Mn^{3+}O_6$ and oxygen defects appeared. Therefore, we consider that the loss of the weight can be ascribed to the oxygen defects.

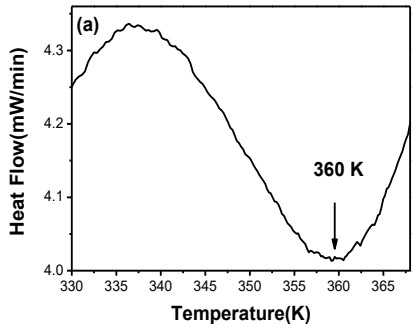 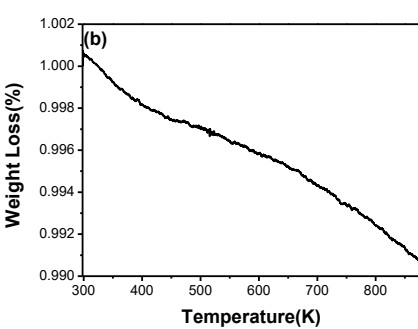

**Figure 9.** DSC (**a**) and TGA (**b**) curve of $Sm_{0.85}Sr_{0.15}MnO_{3-\delta}$.

Moreover, the non-stoichiometric ratio of $Sm_{0.85}Sr_{0.15}MnO_{3-\delta}$ caused the mole ratio mismatch of the Sm, Mn, and O atoms. Some lattice vacancies and interstitials existed in the crystal lattice, making the lattice distortion continuous. In the structure analysis, the crystal distortion was found to have a direct impact on the bond length and angle of the $MnO_6$ octahedron. As for $ABO_3$, when we conducted the substitution in the A position with a different ionic radius, especially in the non-stoichiometric ratio manganese perovskite, the size mismatch effects of the A position ion together with lattice space and interstitial caused a difference in the crystal structure. These eventually led to a great change in the lattice parameters and unit cell size [30–32].

## 4. Conclusions

(1) A novel negative thermal expansion material composed of $Sm_{0.85}Sr_{0.15}MnO_{3-\delta}$ was synthesized using the solid-state method with an NTE coefficient of $-10.08 \times 10^{-6}/K$ from 360 to 873 K.

(2) The particles were homogenous spherical or elliptic–spherical particles with a uniform particle size of about 1~2 μm.

(3) The ceramic $Sm_{0.85}Sr_{0.15}MnO_{3-\delta}$ crystallized in an orthorhombic structure with the space group Pbnm. When $Sr^{2+}$ substituted the $Sm^{3+}$ in $SmMnO_3$, $Sr^{2+}$ occupied the position of $Sm^{3+}$. To maintain the valence balance, electronic transfer occurred in the $Mn^{3+}$, converting into $Mn^{4+}$ in $Sm_{0.85}Sr_{0.15}MnO_{3-\delta}$. The $Mn^{3+}\text{-}O^{2-}\text{-}Mn^{4+}$ structure formed in the process.

(4) The thermal property of $Sm_{0.85}Sr_{0.15}MnO_{3-\delta}$ is considered to be related to the interaction of the lattice vibration and electron transfer between Mn ions. As the temperature rise, the lattice vibrated dramatically and more $Mn^{3+}$ converted into



$Mn^{4+}$. Additionally, the electron transfer rate increased between the $Mn^{3+}$ and $Mn^{4+}$ ions as the temperatures increased. The number of $Mn^{3+}$ ions that can cause the Jahn–Teller effect increasesd. The oxygen ions in the $Mn^{3+}O_6$ octahedron became slant or even produced oxygen defects. The contributions of the lattice vibrations and electron transfer between $Mn^{3+}$ and $Mn^{4+}$ to the thermal expansion changed with the increasing temperature.

(5) The pore energy in the sintered body partially absorbed the expansion of the a-axis a and the c-axis; the negative expansion phenomenon can be explained from the perspective of the contraction of the b-axis. The abnormal thermal expansion behavior of the $Sm_{0.85}Sr_{0.15}MnO_{3-\delta}$ perovskite system is caused by the presence of pores in the sintered body combined with the negative expansion of the b-axis in the perovskite system.

**Author Contributions:** Conceptualization, Y.L. (Yucheng Li); Methodology, Y.L. (Yucheng Li); Data curation, Y.L. (Yucheng Li); software, Y.L. (Yucheng Li); validation, Y.L. (Yongtian Li); formal analysis, Y.L. (Yucheng Li); investigation, Y.Z.; resources, Y.Z.; writing—original draft preparation, Y.Z.; writing—review and editing, Y.W.; visualization, Y.W.; supervision, Y.L. (Yongtian Li); project administration, Y.Z. All authors have read and agreed to the published version of the manuscript.

**Funding:** This research received no external funding.

**Institutional Review Board Statement:** Not applicable.

**Informed Consent Statement:** Informed consent was obtained from all subjects involved in the study.

**Data Availability Statement:** The study did not report any data.

**Conflicts of Interest:** The authors declare no conflict of interest.

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
