# Peer review of "Negative Thermal Expansion Properties of Sm0.85Sr0.15MnO3-δ"

_jcs, doi:10.3390/jcs6060156_

Round 1
Reviewer 1 Report
The authors report the synthesis and characterization of a new material, a 0.85-0.15 solid solution between SmMnO3 and SrMnO3. They observe a fairly large, reversible NTE effect above 360 K. The authors have performed a significant amount of experimental work to determine the origins of NTE in this material, but I'm not fully convinced by their explanation.
Firstly, the English must be improved considerably in order for the manuscript to be clearly understandable. It is possible that I do not fully understand some of the authors' points because of this.
The authors ascribe the observed NTE to a mixture of factors: thermally induced changes in the oxidation state of Mn, concomitant creation of oxygen vacancies, vibrational effects, and microstructural effects. While it is possible or even likely that all of these factors can contribute in some way, none of these factors are studied in great detail.
The evidence for the formation of oxygen vacancies in the structure seems to be limited to TGA data, where a small mass loss on heating could also be explained by the burning off of organic contaminants. The authors should consider performing cyclic TGA experiments to demonstrate reversible formation/loss of oxygen vacancies. The XPS data in Figure 6 is perhaps intended to further this point but is not discussed in the text.
The authors do not provide any evidence for the role of lattice vibrations in NTE in this material, and beyond demonstrating the presence of pores in the structure do not provide evidence of microstructural effects contributing to NTE either. As the dilatometric heating/cooling curves are fairly identical, such microstructural effects are unlikely to be due to crack formation and healing, and likely would require consideration of elastic anisotropy to explain.
As the conclusions are insufficiently supported by the data, I cannot recommend this manuscript for publication.
Reviewer 2 Report
Authors investigatre the negativity of thermal expansion for Sm0.85Sr0.15MnO3-δ. The approach is experimental, and rather logical/traditional. Athough the topic of negative thermal expansion is quite mature, there is sufficient merit to publish this paper based on its correctness.
Author Response
Thanks for the valuable advices given by the reviewer, I will continue to work hard in the future.
Reviewer 3 Report
Paper is interesting, results important form sicentific reason and also possible application of materials with the negative thermal expension. However, there paper needs to be corrected.
Page 6 – there is: „it's easy to have a flexible hole, the pore size becomes larger after gas adsorption”- the „flexible hole” it sholud be explained
Page 9 There is the sentence „ The system energy remains unchanged all the time. This process is known as double-exchange.” – the reference is needed
The same reference is needed in next sentence „However, according to the theory of Zener,…”
Next sentence „The number of Mn3+ that can cause Jahn-Teller effect increases.” also reference to the Jahn-Teller effect is necesary
There are much more information in the paper which needed reference
How the average size of particles (1-2 µm ) was measured?
There is a statement in paper „Therefore, we believe that when the temperature rises, the crystal squeezes the open space, namely these pores and microcracks, which is another reason of negative thermal expension” but first there must be clearly described the main reason of negative thermal expension , it should be underlined in the text.
In sentence „Table 2 are Pore size、Pore volume and BET Surface Area of sample. „ Pore size, Pore volume, Surface Area can be written in smal letter. Moreover, there is no information about the method of measured these values.
Some sentences – English need to be corrected.
Round 2
Reviewer 1 Report
The authors have included some useful information in their reply (in particular that they have demonstrated the reversible formation of oxygen vacancies through TGA). However the manuscript has not been modified accordingly and therefore I cannot recommend it to be accepted.
Reviewer 3 Report
Paper is ready to by published.
This manuscript is a resubmission of an earlier submission. The following is a list of the peer review reports and author responses from that submission.